# Psychometric Properties of the Translated Spanish Version of the Vaginal Penetration Cognition Questionnaire: A Preliminary Work for Validation

**DOI:** 10.3390/healthcare11101482

**Published:** 2023-05-19

**Authors:** Aida Lopez-Brull, Borja Perez-Dominguez, Sergio Hernandez-Sanchez, Alvaro Manuel Rodriguez-Rodriguez, Irmina Nahon, Maria Blanco-Diaz

**Affiliations:** 1Department of Physiotherapy, University of Valencia, 46010 Valencia, Spain; lobrull@alumni.uv.es; 2Exercise Intervention for Health Research Group (EXINH-RG), Department of Physiotherapy, University of Valencia, 46010 Valencia, Spain; 3Department of Pathology and Surgery, Physiotherapy Area, Center for Translational Research in Physiotherapy, Miguel Hernandez University, 03202 Elche, Spain; sehesa@umh.es; 4Department of Physiotherapy, University of Oviedo, 33003 Oviedo, Spain; alvaro.manuel.rodriguez@gmail.com (A.M.R.-R.); blancomaria@uniovi.es (M.B.-D.); 5Discipline of Physiotherapy, Faculty of Health, University of Canberra, Canberra 2617, Australia; irmina.nahon@canberra.edu.au

**Keywords:** psychometrics, reliability, Spain, translation, Vaginal Penetration Cognition Questionnaire, validity

## Abstract

(1) Background: To develop an instrument in Spanish to assess beliefs and feelings about vaginal penetration and assess its psychometric properties. (2) Methods: This study translated and adapted the Vaginal Penetration Cognition Questionnaire into Spanish, and a total of 225 women who suffered from Genito-Pelvic Pain/Penetration Disorder were included in the study. The psychometric properties, including construct, convergent and discriminant validity, test–retest reliability, and internal consistency of the translated version were assessed. (3) Results: The Spanish version of the Vaginal Penetration Cognition Questionnaire is a valid, reliable, and consistent tool to assess beliefs and thoughts about vaginal penetration in women suffering from Genito-Pelvic Pain/Penetration Disorder. The exploratory factor analysis yielded four domains that explained 62.5% of the variance. Convergent and discriminant validity was also confirmed. Test–retest reliability was high, with an intraclass correlation coefficient value of 0.90, a standard error of measurement of 4.21, and a minimal detectable change of 11.66 points. Every domain also showed good internal consistency levels, with Cronbach’s α values ranging from 0.84 to 0.89. (4) Conclusion: The Spanish version of the Vaginal Penetration Cognition Questionnaire is a valid, reliable, and consistent tool to assess vaginal penetration cognition in women suffering from Genito-Pelvic Pain/Penetration Disorder.

## 1. Introduction

Pain during intercourse is a common female sexual dysfunction, classically diagnosed either as “vaginismus”, “vulvodynia”, or “dyspareunia” [1]. The American Diagnostic and Statistical Manual of Mental Disorders, in its 5th edition (DSM-V) integrates vaginismus, vulvodynia, and dyspareunia under a unified term, Genito-Pelvic Pain/Penetration Disorder (GPPPD) [2], a multifactorial condition influenced by biological, psychological, and social factors that generates persistent pelvic pain during sexual intercourse or anticipated vaginal penetration.

Women with GPPPD often have negative beliefs about pain and sexuality, and these may result in avoidance of sexual intercourse and anticipatory anxiety, which ultimately leads to partner avoidance [3]. Beliefs and cognitions about vaginal penetration have also been proposed to play a relevant role in the aetiology, exploratory models, and treatment of sexual pain disorders [4]. For instance, catastrophic pain thoughts, such as “this pain will be unbearable”, subsequently generate a response in the body that increases pelvic muscle tone or vaginal dryness, enhancing the difficulty for penetration.

Despite the potential relevance of beliefs about vaginal penetration in the assessment and treatment of GPPPD, the availability of developed and validated instruments to assess this construct is limited. Usually, the assessment of this construct is contained within instruments that assess several dimensions along with beliefs about vaginal penetration. For instance, instruments such as the Multidimensional Vaginal Penetration Disorder Questionnaire (MVPDQ) [5] and its partner version, Partner Version of the Multidimensional Vaginal Penetration Disorder Questionnaire (PV-MVPDQ) [6], include items involving attitudes and beliefs about vaginal penetration, but they do not assess these per se and are considered substantially long [7]. A shorter, more specific alternative can be found in the Vaginal Penetration Cognitions Questionnaire.

The Vaginal Penetration Cognition Questionnaire (VPCQ) is a 22-item instrument originally developed in English that has been previously validated [4] and translated into other languages, such as Turkish [8] and Persian [9]. However, to the best of our knowledge, a validated translated Spanish version of the VPCQ is lacking, limiting its use by clinicians in Spanish-speaking communities. The VPCQ includes items that are related to control, catastrophizing thoughts, pain, self-image, and genital incompatibility, and includes positive thoughts in its assessment. For this reason, it is a comprehensively complete instrument that assesses beliefs about vaginal penetration. The aim of this study was to translate the VPCQ in Spanish and assess its psychometric properties within a population of women suffering from GPPPD.

## 2. Materials and Methods

### 2.1. Participants

This cross-sectional study was conducted between November 2022 and January 2023. Members of the research team, through their social network accounts and the communication channels of the Faculty of Physiotherapy of the University of Valencia and the Chartered Society of Physiotherapists of Valencia, recruited potential participants through online surveys. A convenience sample of women suffering from GPPPD was recruited. Participants were included if they met the following criteria: (1) adult women (>18 years old), (2) pain episodes in the genital area before, during or after vaginal intercourse for a period lasting at least 3 months, and (3) able to read and write in Spanish. Participants were excluded if they suffered any medical history of other pathophysiological conditions such as infection, gynaecological cancer, vulvar dermatological conditions, painful bladder syndrome, pregnancy, or pelvic surgery, and if they were unable to understand Spanish. Participants were briefed on the study purpose and gave written consent to participate. Ethics approval was obtained from the Ethical Committee in Human Research at the University of Valencia, Spain (UV-INV_ETICA-2557852).

### 2.2. Measures

Baseline demographic information from the sample was obtained, including age, marital status, level of education, and pain duration.

The Vaginal Penetration Cognition Questionnaire is a self-reported instrument designed to assess the thoughts and feelings of women with sexual dysfunctions regarding vaginal penetration. Its original English version included 22 items [4]. It can be answered using a Likert-style scale, ranging from 0 (not at all applicable) to 6 (very strongly applicable). This instrument measures five different aspects of vaginal penetration, classified in subscales, including (1) lack of control of one’s own body and situation during penetration, (2) negative and catastrophic cognitions about future vaginal penetration and expectation of pain, (3) negative cognitions about self-image related to vaginal penetration, (4) positive cognitions, and (5) cognitions regarding genital incompatibility. Scores in each subscale range from 0 to 30 points. Higher scores in positive cognitions indicate better sexual functioning and higher scores in the rest of the subscales indicate higher sexual impairment.

### 2.3. Procedures

#### 2.3.1. Translation and Cultural Adaptation

Before starting the translation and cross-cultural adaptation process, a member of the research team in charge of developing the original English version (Dr. Moniek M. Ter Kuile) was contacted and granted us permission to translate and adapt the VPCQ into Spanish. For a questionnaire to be used in a new language and cultural setting, a comprehensive translation and cultural adaptation must be followed according to established guidelines [10]. The translation process was conducted according to the following steps: (1) The first author (BPD), a native Spanish speaker with fluent English level, translated every item from its original English version to Spanish, and (2) this translation was checked by another author (ALB) who is also fluent in both languages, and any differences in the translation process between both authors were resolved by consensus. Finally, (3) another author (JC), who is also fluent in both languages, back translated the Spanish version to English again. The research team approved the final version of the translated questionnaire by consensus.

#### 2.3.2. Data Collection

Data was collected by a member of the research team and was registered in a separate spreadsheet. Participants were contacted online through a survey created in Google Forms, containing a first block of questions involving their demographic data and a second block of questions containing the VPCQ in Spanish, and responses to that survey were accessed by this researcher. To assess test–retest reliability, a subset of 89 participants who originally completed the online survey was contacted and asked to complete the survey a second time, a week after the completion of the first survey.

### 2.4. Statistical Analysis

Statistical analyses for validity, test–retest reliability, and internal consistency were performed using the SPSS Statistics (IBM, Armonk, NY, USA) software, in its 22.0 version for MacOS. Baseline demographic information was described as means (standard deviations) for continuous data, and counts (percentages) for categorical data. Normal distribution of data was checked through the Kolmogorov–Smirnov test. Missing values were resolved through the imputation method, by replacing them with the participant’s average response.

#### 2.4.1. Sample Size Estimation

Sample size calculation was performed according to published guidelines that establish requirements for survey tool validation (10 participants per item in the tool) [11]. Additionally, the consensus-based standards for the selection of health status measurement instrument (COSMIN) recommendations were followed to perform a structural validity analysis in a sample with an appropriate number of participants, establishing a minimum sample of seven times the number of items and ≥100 [12]. The VPCQ is a 22-item questionnaire, therefore a sample of at least 220 participants was required.

#### 2.4.2. Validity

The validity of the VPCQ was assessed by using an exploratory factor analysis (EFA), and by determining its convergent and discriminant validity [13]. EFA was conducted through an exploratory principal component analysis (PCA) with Varimax rotation, and the number of suitable domains was extracted using the Scree-test criteria [14]. Sampling adequacy was established with the Kaiser–Meyer–Olkin (KMO) measure at >0.5, and the level of significance was established according to the Bartlett test of sphericity [15,16].

Convergent validity of the VPCQ was assessed using the procedure suggested by Fornell and Larcker [17], i.e., calculating the average variance extracted (AVE) and maximum shared squared variance (MSV). Values of AVE between 0.5 and 0.7 indicate acceptable convergent validity, and values > 0.7 indicate good convergent validity. To confirm discriminant validity, values of the MSV must be lower than those of the AVE [9].

#### 2.4.3. Test–Retest Reliability

Test–retest reliability assesses a measure’s stability over time. A measurement is assessed twice over the same group of people and results from both measurements are correlated. The Intraclass Correlation Coefficient (ICC) (model-alpha, 2-way random effects model) was used in this study, establishing scores that ranged from 0 to 1, with 0 being no correlation at all and 1 the highest level of possible correlation. Based on published guidelines [18], scores ranging from 0 to 0.4 indicate low correlation, from 0.4 to 0.6 indicate moderate correlation, from 0.6 to 0.8 indicate average correlation, and scores above 0.8 indicate high correlation levels. Additionally, a Bland–Altman graphical representation was constructed by plotting the mean differences of both measurements with their corresponding limits of average difference ± standard deviation of the difference [19].

Measurement error was also assessed by calculating the standard error of measurement (SEM) and the minimal detectable change (MDC). SEM was calculated using the formula SD × √(1 − R), where SD is the Standard Deviation and R is the reliability coefficient of the instrument [20]. MDC was calculated with the following formula: 1.96 × √2× SEM.

#### 2.4.4. Internal Consistency

Internal consistency refers to the level of interrelationship of the items of a measurement among themselves. Cronbach’s α was used to assess the internal consistency in this study, values ranging between 0 and 1, and the closer to 1 the values are, the better internal consistency. Based on published guidelines [21], Cronbach’s α values > 0.7 indicate acceptable internal consistency, >0.8 good internal consistency, and >0.9 excellent internal consistency.

## 3. Results

### 3.1. Descriptive Statistics

A final total of 225 women were included in the study. Baseline demographic characteristics of the sample are detailed in Table 1. The mean age of the participants was 31.3 years (SD 6.1), and they had suffered GPPPD for a mean average of 24.2 months (SD 13.1). More than half of the women in the sample were single (55%), and 92% of them had at least a High School education level. The Spanish version of the VPCQ (VPCQ-Sp) scores had a mean of 95.0 (SD 13.3) points.

### 3.2. Validity

Results for the exploratory factor analysis and convergent and divergent validity are shown in Table 2. The KMO value for assessing the adequacy of sampling was 0.71, and Bartlett’s test score was X2 = 1868.08 (*p* < 0.001). The EFA yielded a total of four domains (control cognitions, catastrophic and pain cognitions, self-image cognitions, and positive cognitions), which determined 62.5% of the variance in the VPCQ-Sp items. The first domain, “control cognitions”, explained 8.6% of the variance; the second domain, “catastrophic and pain cognition”, explained 17% of the variance; the third domain, “self-image cognitions”, explained 14.6% of the variance; and the fourth domain, “positive cognitions”, explained 22.3% of the variance.

AVE values were 0.513 for control cognitions, 0.621 for catastrophic and pain cognitions, 0.585 for self-image cognitions, and 0.601 for positive cognitions. All of them were above 0.5, therefore showing an acceptable convergent validity. Additionally, MSV values for every domain were lower than the AVE values; therefore, discriminant validity was also confirmed.

### 3.3. Test–Retest Reliability

Results for test–retest reliability are shown in Table 3. The overall intraclass correlation coefficient for the VPCQ-Sp was 0.90 (95% CI 0.85 to 0.93). Out of the total of 22 items, 20 showed high correlation levels (items 1–7, 9, and 11–22), with correlation values ranging from 0.80 to 0.97. The remaining two items, 8 and 10, showed average correlation levels of 0.75 and 0.78, respectively. The standard error of measurement was 4.21, and the minimal detectable change was 11.66 points. The Bland–Altman plot (Figure 1) shows that most of the pair differences are between the agreement limits, implying a high concordance between the test–retest measures of the VPCQ-Sp.

### 3.4. Internal Consistency

Results for the internal consistency of the domains of the VPCQ-Sp are shown in Table 4. Every domain showed good internal consistency values. Cronbach’s α values for the first domain, “control cognitions”, were 0.89 (95% CI 0.87–0.92); for the second domain, “catastrophic and pain cognition”, 0.84 (95% CI 0.81–0.86); for the third domain, “self-image cognitions”, 0.86 (95% CI 0.83–0.89); and for the fourth domain, “positive cognitions”, 0.84 (95% CI 0.82–0.87).

## 4. Discussion

This is the first study that assesses the psychometric properties of the Spanish version of the VPCQ. Our results show that VPCQ-Sp has high test–retest reliability and excellent internal consistency levels, along with acceptable convergent and discriminant validity. The VPCQ was originally developed by Klaasen and Ter Kuile [4] in English, and they proved it was a reliable and valid instrument to assess cognitions about vaginal penetration in a Dutch population. Furthermore, Dogan et al. [8] and Banaei et al. [9] developed versions translated into Turkish and Persian, respectively.

The above-mentioned studies conducted their analyses in samples of women that suffered from vaginismus and dyspareunia. Our sample was composed of women diagnosed with GPPPD, so although both vaginismus and dyspareunia are conditions considered to be included within the GPPPD spectrum, there was no differentiation of patient diagnosis in this study, as proposed by other authors [22,23]. Despite this, our sample shared similar ages and educational levels to those found in the original [4] and Persian translated version [9] studies, so results are consistent throughout the literature with similar samples. However, to further understand the applicability of the instrument in women with different conditions, further analyses should be explored, and differences between women diagnosed with different conditions should be analysed.

The exploratory factor analysis yielded four domains. This contrasts with the original version, which enhanced the importance of genital incompatibility as a domain per se. The original developers suggest that “vaginismus” should be reconceptualized as either an “aversion or phobia” to vaginal penetration, so the consideration of genital incompatibility as a relevant domain might be linked with this fear-avoidance behaviour rather than with the actual condition [4]. Additionally, our exploratory factor analysis contrasts with the Persian version, as authors conducting this study yielded three domains, combining three domains from the original study (catastrophic and pain cognitions, control cognitions, and genital incompatibility cognitions) into one single domain (catastrophic and control cognition). These differences between versions might be related to the cultural setting in which they are conducted, as there might be subtle language variations or interpretations of some of the items depending on the country in which the items are assessed. Authors of the Persian version even suggest assessing differences within samples from the same country, as multi-cultural countries, such as Iran or Spain, could present different results in different regions.

Validity assessments showed acceptable convergent and discriminant validity levels. The Persian translated version [9] found similar convergent validity results, but a second confirmatory factor analysis was performed because discriminant validity was not confirmed. These differences might also imply the relevance of cultural and social setting when assessing the relatedness of these instruments to their construct, as these thoughts could vary depending on how “taboo” they are perceived to be in a society [24].

Test–retest reliability results are consistent with those found both in the original and translated versions. Every item scored high reliability values except for items 8 and 10. These two items deal with relatively ambiguous concepts such as “being a complete woman” and “being a good partner”; beliefs often influenced by multiple factors which could be a possible explanation for this finding. These items could be interpreted ambiguously, not only between validated versions, but also between assessments within the same sample, as every woman has her own perspective of what “a complete woman” or a “good partner” is.

In this study, good internal consistency values were found. Previous studies have also shown similar values of Cronbach’s α on their subscales, ranging from 0.70 to 0.83 in the original version [4], 0.82 to 0.93 in the Persian version [9], and 0.71 to 0.92 in the Turkish version [8]. However, Banaei et al. also assessed the composite reliability (CR) in their study, arguing this is a much more accurate and reliable assessment because, compared to Cronbach’s α, it is not affected by the number of items and structure of the instrument. This should be taken under account when conducting future internal consistency assessments. Additionally, it should be noted that, to the best of our knowledge, this is the first study to calculate the SEM and the MDC.

The study results must be taken cautiously due to the following limitations. Firstly, even though the translation process was conducted with maximal effort, we do have to consider that the sample in this study is from a different country and cultural setting than that found in similar studies, and that might influence subtle linguistical variations or item interpretations. Another limitation could be that, in this study, there was no registration of the history of treatments participants had received. Participants could have asked for professional help prior to this study, and if they received any kind of therapeutic educational intervention or cognitive therapy, these might have had a significant influence on their answers.

However, this is the first translation of the VPCQ in Spanish, a language spoken by over 548.3 million speakers around the world, and the native language of over 20 countries [25]. Therefore, the development of the Spanish version of the VPCQ could imply a relevant addition for so many clinicians and researchers. Adding validated instruments to the complex diagnosis of women suffering from these conditions could bring clinicians closer to a more effective therapeutic approach.

Finally, this is a preliminary validation study, as further statistical analyses could have been performed. This study serves as the first steppingstone towards a complete validation process. Future studies should focus on conducting a confirmatory factor analysis on a different sample, an analysis of the invariance with respect to the original instrument, and a correlation analysis between the VPCQ and a similar validated instrument in Spanish. This last analysis could be challenging, as there are very few validated instruments that assess a similar construct, so initial correlational analyses should focus on constructs that are as close as possible, such as sexual function or pain cognitions.

## 5. Conclusions

This is a preliminary study that translated and culturally adapted the Vaginal Penetration Cognition Questionnaire in Spanish. The Spanish version of the Vaginal Penetration Cognition Questionnaire is a valid, reliable, and consistent tool for assessing vaginal penetration cognitions in women suffering from Genito-Pelvic Pain/Penetration Disorder.

## Figures and Tables

**Figure 1 healthcare-11-01482-f001:**
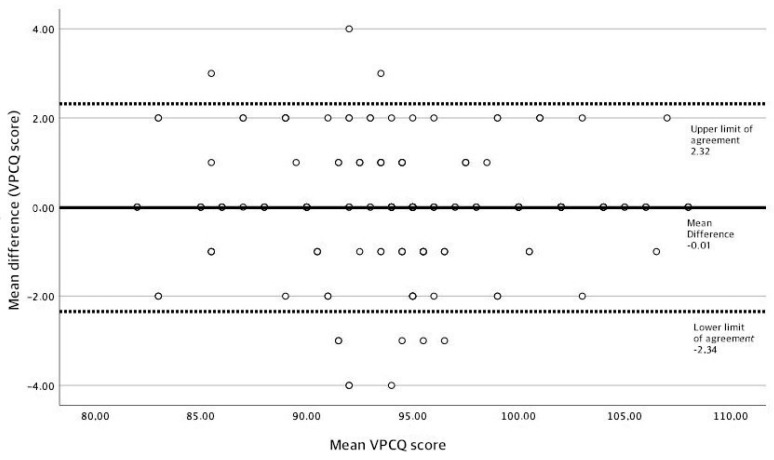
Bland–Altman plot VPCQ total score.

**Table 1 healthcare-11-01482-t001:** Baseline demographic characteristics from the sample.

Outcome		Mean (SD) or n (%)
Age (years)		31.3 (6.9)
Pain duration (months)		24.2 (13.1)
Marital status	Married	83 (37%)
	Divorced	15 (7%)
	Separated	3 (1%)
	Widowed	1 (0%)
	Single	123 (55%)
Educational level		
	Uneducated	0 (0%)
	Primary	18 (8%)
	High School	116 (52%)
	College	91 (40%)

SD: Standard Deviation.

**Table 2 healthcare-11-01482-t002:** Exploratory factor analysis results, convergent and discriminant validity of the Spanish version of the VPCQ.

Domain	Mean (SD)	Loading	Variance (%)	AVE	MSV
1. Control cognitions			8.6	0.513	0.387
Item 2. Tengo miedo de perder el control sobre la situación durante la penetración	5.1 (1.2)	0.692			
Item 6. Tengo miedo de entrar en pánico durante la penetración	4.5 (2.3)	0.809			
Item 20. Tengo miedo de no poder influir sobre lo que sucede durante la penetración	5.0 (1.4)	0.718			
Item 21. No saber qué ocurre en mi cuerpo durante la penetración me da miedo	5.7 (0.3)	0.662			
2. Catastrophic and pain cognitions			17.0	0.621	0.515
Item 5. Lo más seguro es que la penetración no sea posible	4.9 (1.2)	0.733			
Item 7. Tengo miedo de que la penetración se vuelva más difícil en un futuro	5.1 (1.8)	0.660			
Item 9. Tengo miedo de que el dolor en la penetración vaya a más en un futuro	5.3 (0.9)	0.666			
Item 11. Seguramente la penetración no se podrá realizar	5.6 (1.4)	0.693			
13. El pene de mi pareja es demasiado grande para mi vagina	3.1 (2.3)	0.527			
Item 16. Tengo miedo de sentir calambres durante la penetración	4.5 (0.8)	0.748			
3. Self-image cognitions			14.6	0.585	0.515
Item 1. Tengo miedo de que mi vagina sea demasiado estrecha para la penetración	4.5 (0.7)	0.497			
Item 8. Sólo me siento una “mujer completa” cuando la penetración es posible	2.1 (1.3)	0.707			
Item 10. Soy una mala pareja cuando la penetración no es posible	2.5 (1.0)	0.715			
Item 15. Soy la única en el mundo con la que la penetración no es posible	2.8 (1.5)	0.524			
Item 17. Me siento culpable cuando la penetración no es posible	5.5 (0.9)	0.635			
Item 19. Tengo miedo de que mi pareja me deje porque la penetración no es posible	3.8 (2.7)	0.786			
4. Positive cognitions			22.3	0.601	0.301
Item 3. Siento que la penetración será agradable	4.0 (1.6)	0.744			
Item 4. La penetración es un momento de intimidad con mi pareja	5.3 (0.5)	0.623			
Item 12. Me excitaré sexualmente con la penetración	4.0 (1.3)	0.762			
Item 14. La penetración será agradable	4.2 (2.1)	0.857			
Item 18. La penetración acabará en un orgasmo	4.8 (1.3)	0.613			
Item 22. Aunque la penetración no sea posible, soy una buena pareja sexual	3.6 (1.3)	−0.663			

**Table 3 healthcare-11-01482-t003:** Intraclass correlations of the items in the Spanish version of the VPCQ.

Item	Description	ICC	95% CI	SEM	MDC
Total		0.90	0.85–0.93	4.21	11.66
1	Tengo miedo de que mi vagina sea demasiado estrecha para la penetración	0.85	0.74–0.94		
2	Tengo miedo de perder el control sobre la situación durante la penetración	0.81	0.78–0.84		
3	Siento que la penetración será agradable	0.80	0.71–0.88		
4	La penetración es un momento de intimidad con mi pareja	0.95	0.88–0.97		
5	Lo más seguro es que la penetración no sea posible	0.92	0.87–0.95		
6	Tengo miedo de entrar en pánico durante la penetración	0.86	0.80–0.89		
7	Tengo miedo de que la penetración se vuelva más difícil en un futuro	0.86	0.82–0.93		
8	Sólo me siento una “mujer completa” cuando la penetración es posible	0.75	0.68–0.79		
9	Tengo miedo de que el dolor en la penetración vaya a más en un futuro	0.95	0.92–0.97		
10	Soy una mala pareja cuando la penetración no es posible	0.78	0.70–0.81		
11	Seguramente la penetración no se podrá realizar	0.90	0.86–0.93		
12	Me excitaré sexualmente con la penetración	0.93	0.91–0.96		
13	El pene de mi pareja es demasiado grande para mi vagina	0.85	0.73–0.95		
14	La penetración será agradable	0.93	0.89–0.95		
15	Soy la única en el mundo con la que la penetración no es posible	0.82	0.79–0.83		
16	Tengo miedo de sentir calambres durante la penetración	0.94	0.92–0.98		
17	Me siento culpable cuando la penetración no es posible	0.92	0.90–0.95		
18	La penetración acabará en un orgasmo	0.90	0.87–0.92		
19	Tengo miedo de que mi pareja me deje porque la penetración no es posible	0.82	0.79–0.84		
20	Tengo miedo de no poder influir sobre lo que sucede durante la penetración	0.95	0.93–0.96		
21	No saber qué ocurre en mi cuerpo durante la penetración me da miedo	0.97	0.95–0.98		
22	Aunque la penetración no sea posible, soy una buena pareja sexual	0.88	0.85–0.90		

**Table 4 healthcare-11-01482-t004:** Internal consistency coefficients for the factors of the Spanish version of the VPCQ.

Domain	Cronbach’s α (95% CI)
1. Control cognitions	0.89 (0.87–0.92)
2. Catastrophic and pain cognitions	0.84 (0.81–0.86)
3. Self-image cognitions	0.86 (0.83–0.89)
4. Positive cognitions	0.84 (0.82–0.87)

## Data Availability

Data will be made available upon request by the Corresponding author.

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
