# Peer review of "Psychometric Properties of the Translated Spanish Version of the Vaginal Penetration Cognition Questionnaire: A Preliminary Work for Validation"

_healthcare, 2023, doi:10.3390/healthcare11101482_

Round 1
Reviewer 1 Report
This article proposes to apply validation in the Spanish context of an instrument of great scientific relevance (Vaginal Penetration Cognition Questionnaire).
Notwithstanding this aim, the article could only be defined as a preliminary work in the complex process of validation, as it lacks certain fundamental elements for full and final validation.
For example, a confirmatory factorial analysis (CFA) was not carried out on a different sample.
Furthermore, the results of this work were not correlated with those of at least one other standardised and validated assessment instrument in the Spanish context.
For these reasons, the present work could, in my opinion, only be published as a "preliminary work for validation", which should be followed by a further article (or research step) in which the confirmatory factor analysis is applied to another group of participants.
In the following research step, the results obtained from the present instrument have to be correlated with those obtained simultaneously with another instrument validated in Spain (inquiring about related domains such as sexual functionality, pain on penetration, and so on).
In my opinion, the work could be published in this form, as long as it is made explicit that this is preliminary data for validation, which will be followed by subsequent work to evaluate the further psychometric characteristics of the instrument as listed above.
Author Response
Authors would lilke to thank the Reviewer for the analysis and feedback. We agree with the Reviewer that a more thorough analysis should be done in a validation process, and we understand that further analyses should've been done because of this.
It was challenging to correlate the results from our analysis with those of another validated assessment instrument that assessed the same or similar construct, as there are not many. We agree with the Reviewer that including "a preliminary work for validation" in the title could enable a future study that fullfils a more thorough analysis.
We take under consideration the Reviewer's advice, and we will consider a further analysis to complete the validation process of this article.
Again, authors would like to thank the Reviewer for the feedback.
Reviewer 2 Report
This is a well written manuscript on the validity of the Spanish version of the Vaginal Penetration Cognition Questionnaire. I think that the authors did a good job in translating the items into Spanish using back translation. However I do have two concerns about the statistical analysis.
1. The authors rely on exploratory factor analysis to test the factor structure of the translated questionnaire. This is a suboptimal choice as there is a clear expectation about the structure of the questionnaire. That is, the questionnaire should follow the same structure as the original questionnaire. In such situations, confirmatory factor analysis is a much more suitable approach as it provides a means to test the theoretical structure in a statistically falsifiable way. As a result, it can be more rigorously studied if the translated questionnaire follows the desired theory with which the questionnaire was originally developed.
2. Demonstrating that the translated questionnaire follows the theoretical factor structure is a necessary but not a sufficient condition for the translated questionnaire to be valid. A translated questionnaire should be demonstrated to have measurement invariance with respect to the original questionnaire (see e.g., Ellis, 1989). This is equivalent to demonstrating that the translated items do not show differential item functioning. In testing this, it can be studied if the questionnaire measures the same psychometric constructs as the original questionnaire. Violations of measurement invariance indicate that the corresponding items are not performing as intended. The authors can test for measurement invariance between their sample and the normalization sample of the original questionnaire (or any other sample that is administered the original questionnaire, or a validated translation).
Ellis, B. B. (1989). Differential item functioning: Implications for test translations. Journal of applied psychology, 74(6), 912.
Author Response
Authors would like to thank the Reviewer for the comments and feedback. We understand the concerns involving the analysis we've performed.
We agree that a confirmatory factor analysis would've been a more suitable choice to test the factor structure. We added "a preliminary work of validation" to our title, as the process of validation could be further developed including the suggested analysis by the Reviewer.
Authors would also like to thank the Reviewer for the suggestion of measuring the invariance with respect to the original instrument. Again, we agree that adding this analysis to our validation process would've improven the robustness of our study, so we understand that this study could serve as baseline to develop further studies that include these analyses.
We understand that not including these thorough analyses in our study is a limitation that directly affects the robustness of our results and conclusions. We agree that researchers conducting these types of analysis should take these advices under consideration.
Again, we would like to thank the Reviewer for the suggestions and for the provided feedback.
Round 2
Reviewer 1 Report
The clarifications included in the review by the authors provide a better definition of the characteristics of the study. In my opinion, it can now be published in this version.